# Enabling Continuous Wearable Reflectance Pulse Oximetry at the Sternum

**DOI:** 10.3390/bios11120521

**Published:** 2021-12-17

**Authors:** Michael Chan, Venu G. Ganti, J. Alex Heller, Calvin A. Abdallah, Mozziyar Etemadi, Omer T. Inan

**Affiliations:** 1Wallace H. Coulter Department of Biomedical Engineering, Georgia Institute of Technology, Atlanta, GA 30332, USA; mchan81@gatech.edu (M.C.); calvinabdallah@gatech.edu (C.A.A.); 2Bioengineering Graduate Program, Georgia Institute of Technology, Atlanta, GA 30332, USA; vganti6@gatech.edu; 3Department of Anesthesiology, Feinberg School of Medicine, Northwestern University, Chicago, IL 60611, USA; james.heller@northwestern.edu (J.A.H.); mozzi@northwestern.edu (M.E.); 4Department of Biomedical Engineering, McCormick School of Engineering, Northwestern University, Evanston, IL 60201, USA; 5School of Electrical and Computer Engineering, Georgia Institute of Technology, Atlanta, GA 30332, USA

**Keywords:** reflectance pulse oximetry, oxygen saturation, photoplethysmogram (PPG), continuous monitoring, respiratory monitoring, outlier rejection

## Abstract

In light of the recent Coronavirus disease (COVID-19) pandemic, peripheral oxygen saturation (SpO_2_) has shown to be amongst the vital signs most indicative of deterioration in persons with COVID-19. To allow for the continuous monitoring of SpO_2_, we attempted to demonstrate accurate SpO_2_ estimation using our custom chest-based wearable patch biosensor, capable of measuring electrocardiogram (ECG) and photoplethysmogram (PPG) signals with high fidelity. Through a breath-hold protocol, we collected physiological data with a wide dynamic range of SpO_2_ from 20 subjects. The ratio of ratios (R) used in pulse oximetry to estimate SpO_2_ was robustly extracted from the red and infrared PPG signals during the breath-hold segments using novel feature extraction and PPG_green_-based outlier rejection algorithms. Through subject independent training, we achieved a low root-mean-square error (RMSE) of 2.64 ± 1.14% and a Pearson correlation coefficient (PCC) of 0.89. With subject-specific calibration, we further reduced the RMSE to 2.27 ± 0.76% and increased the PCC to 0.91. In addition, we showed that calibration is more efficiently accomplished by standardizing and focusing on the duration of breath-hold rather than the resulting range in SpO_2_. The accurate SpO_2_ estimation provided by our custom biosensor and the algorithms provide research opportunities for a wide range of disease and wellness monitoring applications.

## 1. Introduction

Due to the novel Coronavirus disease (COVID-19) pandemic, there is a clear need to monitor respiratory functions in outpatient settings to help assess the progression of COVID-19 during the presymptomatic, symptomatic, and recovery stages. In a recent effort to record and model the trajectories of several vital signs in hospitalized COVID-19 patients, Pimentel et al. showed that peripheral oxygen saturation (SpO_2_) is amongst the most indicative of parameters of COVID-19 progression prior to primary outcomes, suggesting the importance to monitor SpO_2_ continuously [1]. Through remote SpO_2_ monitoring, accurate tracking of COVID-19 progression allows for the implementation of disease-management strategies for both timely interventions and the optimization of scarce medical resources [2].

Unfortunately, existing SpO_2_ measurement devices are inconvenient for monitoring in outpatient settings. Typically, SpO_2_ is measured through pulse oximeters placed at peripheral extremities such as the fingers; however, these devices obstruct normal activities of daily living (ADLs) due to restriction of finger usage. In addition, finger-clip based pulse oximeters are accordingly limited in practice to intermittent or single-point measurements. Recently, commercial wrist-worn devices such as Apple Watch, Fitbit Sense Advanced Smartwatch, and Garmin Vivosmart 4 have been developed that allow for more convenient monitoring and offer continuous SpO_2_ measurements. Unfortunately, SpO_2_ measured at the wrist is likely more susceptible to motion artifacts when compared to measurement sites closer to the center of mass of the body, such as the forehead and ear during walking, as shown by Longmore et al. [3]. In addition, peripheral sites such as the wrist respond to apnea events at a slower than central sites [4,5] due to the redistribution of blood flow, oxygen conservation [6], and their distal location to the heart.

Thus, central pulse oximeters offer a promising approach for ambulatory outpatient monitoring and in detecting acute hypoxemia events. Furthermore, chest-based pulse oximeters might be advantageous for their synergistic incorporation with other cardiac monitoring methods—such as electrocardiography and seismocardiography—for a more holistic understanding of cardiac functions [7,8,9,10]. Nevertheless, a few major challenges such limitations to the reflectance design, the presence of respiratory artifacts, and the malperfusion of the sternum pose difficulties to the adoption of chest-based approaches. Despite these challenges, prior evaluations have suggested that a chest-based approach might be feasible [3,11,12,13,14,15].

However, existing methods using chest-based devices are not rigorously validated, and thus more work is needed to advance chest-worn pulse oximetry. Specifically, validating chest-based pulse oximetry for continuous monitoring would require a sufficiently large and diverse subject population, a wide dynamic range of SpO_2_, a resultant root-mean-square error (RMSE) lower than 3.5% [16], and ideally a form factor that does not interfere with ADLs. Upon investigation, there are some clear gaps in the existing literature in these areas that should be addressed. For example, either insufficient sample size (N = 1) in [11] or narrow dynamic range for SpO_2_ (88–99%) in [12] limits the validity of the accuracy of their approaches. Meanwhile, Näslund et al. [15] showed a strong agreement between their estimated SpO_2_ and arterial oxygen saturation (SaO_2_); however, their device was unable to capture the pulsatile component of the PPG signals and therefore might be susceptible to motion artifacts and skin pigmentation [17] (p. 34). Kramer et al. [13] achieved accurate SpO_2_ estimations (RMSE of 2.9%, N = 13), but their work lacks the key details necessary for replicating their algorithm such as preprocessing and feature extraction. Finally, Vetter et al. [14] conducted a study following the International Organization for Standardization 9919 international standard, included a moderate number of subjects (N = 10) and a wide dynamic range of SpO_2_ (70–99%), and provided sufficient details of their approach. Nevertheless, the prototype appears to be cumbersome to use as it requires a chest-strap to affix the device for sufficient contact pressure. Additionally, due to the lack of leave-one-subject-out (LOSO) cross validation for training and testing the model, their method is susceptible to data leakage and therefore the high accuracy attained may not be generalizable. To the best of our knowledge, there exists no known accurate chest-based pulse oximetry approach that has been thoroughly validated within the literature. This work attempts to address these shortcomings and present a chest-based pulse oximeter that can estimate SpO_2_ in close agreement with a commercially available, validated finger pulse oximeter.

The aim of this paper is to demonstrate the feasibility of our small, standalone chest-based wearable patch biosensor. In this work, we collected data from 20 subjects who underwent a breath-hold perturbation to induce hypoxia and used PPG signals from a custom chest-based biosensor to estimate SpO_2_. To estimate SpO_2_ robustly, we present a novel algorithm to extract key PPG features to account for poor perfusion at the sternum [18] and involuntary respiratory artifacts [19]. In addition, we developed a PPG_green_-based outlier rejection algorithm for rejecting red and infrared (IR) PPG beats of lower quality. Finally, we demonstrate the optimal calibration scheme for practical usage of this chest-based pulse oximetry. These contributions pave the way for the continuous monitoring of SpO_2_. The complete study protocol and algorithmic advancements necessary to achieve accurate chest-based pulse oximetry are described in detail below.

## 2. Materials and Methods

### 2.1. Principle of PPG

The PPG signal used in this work represents the changes in the reflected light emitting diodes’ (LEDs) light intensity, as detected by the photodiodes (PDs). According to the Beer–Lambert law, the intensity of the reflectance PPG measured is related to the optical path length of light traveled from the LEDs to the PDs [17] (pp. 47–48). The changes of PPG intensity with respect to each component (arterial blood, venous blood, tissue, bone, etc.) have different pulsating dynamics [20]. By using appropriate filter banks, we can leverage the cardiac pulsation of the PPG to target its arterial, pulsatile, small-signal component. Specifically, the portion of the PPG signal that is representative of cardiac pulsation and the periodic changes in blood volume is termed the alternating current (AC) component, and the baseline wander of the PPG—which is slower than the cardiac frequency—is termed the direct current (DC) component [20]. The AC and DC components of the PPG in multiple wavelengths (i.e., red and IR PPG) can reveal the oxygenation saturation of the underlying arteries [17]. More details are provided in Section 2.4.

### 2.2. Breath-Hold Study Design

The breath-hold study was designed to induce hypoxemia and sufficient changes in SpO_2_. This study was conducted under a protocol approved by the Georgia Institute of Technology Institutional Review Board (H21100). A total of 22 (16 males, 6 females) young volunteers were recruited for the breath-hold study and written informed consent was obtained. The number of subjects recruited exceeds that of similar studies [14,21,22]. In this dataset, two subjects were excluded for analysis. The data of one subject suffered from poor ECG quality—due to expired ECG electrodes that were inadvertently used. The data of the subject exhibited an abnormal distribution of the extracted features compared to those shown in [17] (p. 51). Specifically, the ratio of ratios (R) systematically deviates more than three standard deviations across all SpO_2_ levels. Therefore, for this work, only data of the remaining 20 subjects were used for analysis. Demographic information of these 20 subjects including age, weight, height, Fitzpatrick skin type, perfusion indices (PI), etc. are summarized in Table 1. Note that the distribution of PI for red and infrared in this dataset falls well below the poor perfusion threshold (0.3%) as defined by the Food and Drug Administration (FDA) [16], suggesting this measurement site is indeed malperfused.

In the breath-hold study, subjects were first asked to shave their chest hair to reduce interference. Subsequently, each subject performed 10 end-expiratory breath-holds while sitting in an upright posture with a one-minute break between breath-holds. One minute was found to be sufficiently long for SpO_2_ to return to its baseline level. Subjects were instructed to hold their breath for as long as possible. Throughout the study, subjects wore a nose clip and held the disposable mouthpiece (AFT36 bacteriological filter; Biopac System Inc., Santa Barbara, CA, USA) between their lips. After the data were collected, important oxygenation/deoxygenation events were manually labeled.

As depicted in Figure 1, we collected the following information: ECG (Biopac ECG100A; Biopac System Inc., Santa Barbara, CA, USA), right index finger SpO_2_ (Biopac OXY100E, TSD124A Finger Clip Transducer; Biopac System Inc.), and respiratory flow (TSD117A Medium Flow Pneumotach Transducer; Biopac System Inc.) data, all sampled at 2000 Hz. The Biopac OXY100E module reports an accuracy ±2% for a SpO_2_ range of 70–100%. We used the 3M™ Red Dot™ ECG electrodes (model 2660; 3M, Saint Paul, MN, USA) throughout the study. Data outside of this SpO_2_ range were discarded since the accuracy is unknown. 

In parallel, we also attached the wearable patch biosensor to the subject’s mid-sternum and collected single-lead ECG, two sets of multiwavelength PPGs (red, infrared [IR], and green), and triaxial seismocardiogram (SCG, not used in this study), sampled at 500, 67, and 1000 Hz, respectively. The hardware used in the biosensor is almost identical to that reported in our previous work [8,10,23] except for the addition of the PPG modules and the change in form factor. The ECG analog front-end (AFE) and the accelerometer AFE (for SCG) remain the same. Specifically, the PPG AFE used to drive the LEDs and obtain data from the PDs is the Maxim 86170 (Maxim Integrated, San Jose, CA, USA). The multi-chip LEDs, which has red (660 nm), and IR (950 nm), and green (526 nm) wavelengths, are the SFH 7016 (OSRAM, Munich, Germany), and the PDs are the VEMD 8080 (Vishay Semiconductors, Heilbronn, Baden-Württemberg, Germany). Serial Peripheral Interface was used as the communication protocol between the microcontroller and peripheral sensors. This device is also equipped with wireless capabilities (i.e., Bluetooth and Wi-Fi) for transmitting data. However, in this study, data were stored in the Secure Digital card and later retrieved by a custom-built software application as in previous work [8,10,23]. The battery life of the device at the full sample rates of all sensors is up to 60 h. The front and lateral views of the device are shown in Figure 2. 

### 2.3. Manual Labeling

In Figure 3, filtered, high-quality physiological signals acquired during breath-hold and breathing are presented. For each subject, we selected the photodiode (PD) with higher quality as determined by visual inspection. The discrepancy can be attributed to the differences in LED/PD separation distance as LED/PD separation distance can affect the quality of PPG [17] (p. 88). Further assessment may be needed since optimizing the LED/PD separation distance is a critical factor for obtaining a good quality signal. Manual labeling was performed using the respiratory flow and the SpO_2_ data. Alignment was necessary since it has been observed that deoxygenation events do not occur simultaneously for different body sites, and SpO_2_ measured at the finger is usually delayed from SpO_2_ measured at central sites [4,5,24]. This delay can be partially attributed to the oxygen-conserving effect induced by breath-hold. Similar to the diving response [6], breath-hold also leads to bradycardia and peripheral vasoconstriction to reduce oxygen consumption in peripheries and redistribute blood flow to vital organs such as the brain and the heart [6]. The combined effect leads to a delayed deoxygenation measurement by a finger-pulse oximeter when compared to a pulse oximeter placed closer to the heart or the brain. Davies et al. reported a mean delay of 16.75 ± 5.88 s across subjects for their in-ear reflectance pulse oximetry [4].

From the respiratory flow (top signal in Figure 3), breathing (the “oscillating” part, pink) and breath-hold (the “silent” part, blue) segments can be easily distinguished. From the ground truth SpO_2_ (the second signal on the left in Figure 3), three distinct timestamps were recorded of each deoxygenation event: start, nadir, and end. The start of deoxygenation is defined as the point where SpO_2_ begins to drop drastically (rate of SpO_2_ decline > 0.5%/cardiac cycle for 3 consecutive cardiac cycle). The nadir of deoxygenation is defined as the lowest SpO_2_ within the deoxygenation event. The end of deoxygenation is defined as the point where SpO_2_ returns to the baseline level. Usually, the nadir and the end of deoxygenation can be easily identified. To account for the delay of deoxygenation between finger and chest deoxygenation, the nadir deoxygenation of the finger is aligned to the end of the breath-hold, based on the assumption that chest arteries received well-oxygenated blood immediately following the end of the breath-hold. Though our results suggest this alignment procedure is somewhat accurate, we found that a more precise alignment algorithm was required to achieve adequate accuracy; the updated alignment algorithm is applied and provided in Appendix A.

### 2.4. Signal Processing Pipeline

#### 2.4.1. Principle of Pulse Oximetry

To relate the aligned signals to ground truth SpO_2_, relevant features in the biosensor signals need to be extracted. We followed the standard approach described in [17] (p. 131) and tailored the algorithm to our pulse oximetry. The key feature, R, defined as the ratio of the normalized AC component (also a ratio) of two optical wavelengths, can be extracted from the PPG signals:(1)R=ACredACIRDCredDCIR 
where AC_red_ is the AC component of the red PPG, AC_IR_ is the AC component of the IR PPG, DC_red_ is the DC component of the red PPG, and DC_IR_ is the AC component of the IR PPG. Normalization is performed by dividing the AC component of a wavelength by its DC component. R, along with the absorption coefficients of oxyhemoglobin (HbO_2_) and deoxyhemoglobin (Hb) for different wavelengths, can be used together to derive SaO_2_ directly. According to [17] (p. 50), the theoretical relationship between SaO_2_ and R is defined as:(2)$${\mathrm{SaO}}_{2}=\frac{\varepsilon_{\mathrm{Hb}}\left(\lambda_{\mathrm{red}}\right)-\varepsilon_{\mathrm{Hb}}\left(\lambda_{\mathrm{IR}}\right)R\;}{\varepsilon_{\mathrm{Hb}}\left(\lambda_{\mathrm{red}}\right)-\varepsilon_{{\mathrm{HbO}}_{2}}\left(\lambda_{\mathrm{red}}\right)+\left[\varepsilon_{{\mathrm{HbO}}_{2}}\left(\lambda_{\mathrm{IR}}\right)-\varepsilon_{\mathrm{Hb}}\left(\lambda_{\mathrm{IR}}\right)\right]R}$$
where ε_Hb_ is the absorption coefficients of Hb, ε_HbO2_ is the absorption coefficients of HbO_2_, λ_red_ is the wavelength of the red PPG, and λ_IR_ is the wavelength of the IR PPG. If further approximated using a Taylor series expansion,
(3)$$\hat{{\mathrm{SaO}}_{2}}=\frac{A\times R+B}{C\times R+D}\approx m\times R+b={\mathrm{SpO}}_{2}$$
emerges as an empirical model that governs the relationship between SpO_2_, the surrogate of SaO_2_, and R. In Equation (3), A, B, C, and D replace the absorbance coefficients of the Hb and HbO_2_ at the two wavelengths [17] (p. 54), *m* is the slope, and *b* is the intercept.

#### 2.4.2. Preprocessing Overview

Figure 4a illustrates our signal-preprocessing pipeline, which we used to extract R from the wearable patch biosensor signals. Although there are other signals in the dataset, we only found six that are relevant to this work, namely, finger SpO_2_, red PPG, IR PPG, green PPG, ECG, and respiratory flow. The Biopac and biosensor signals were first resampled to 500 Hz and synchronized by maximizing the cross-correlation of their ECG signals. The Biopac SpO_2_ was further aligned to the biosensor signals on a per breath-hold basis, using the manual label described in Section 2.3. Due to both respiratory artifacts when emerging from breath-hold and the lack of range in measured SpO_2_ values, we only target SpO_2_ estimation during the breath-hold segments of the signals. Each extracted R from the breath-hold segments was paired with the manually aligned SpO_2_, and both a scatter plot—demonstrating the correlation between R and SpO_2_—and distributions plots to show the skewness of R and SpO_2_—are depicted in Figure 4a. The skewness of R and SpO_2_ can be partially attributed to our ability to maintain oxygenation homeostasis, enabled by the continuous supply of oxygen by the oxygen stores upon breath-hold [6]. Before uncovering the relationship between R and SpO_2_, we will first demonstrate robust feature extraction and the outlier rejection algorithms necessary to extract R for the chest-based pulse oximetry.

#### 2.4.3. Robust Feature Extraction via Linear Transformation

To compute R, it is necessary to compute AC features and DC features of each PPG beat. Note that a feature represents a scalar value to represent the characteristic of a PPG beat in the context of this work. In Figure 4b, the block diagram for beat segmentation and feature extraction has been shown. To isolate the AC component, an empirically validated bandpass filter, with a passband of 0.35 to 4 Hz was first applied. The low cutoff was chosen to remove the baseline wander, due to involuntary respiratory movement. The high cutoff was chosen empirically so as to reduce the dicrotic notch and preserve only the frequency components with less variation across wavelengths. The AC component was segmented into PPG beats using ECG R-peaks detected by a Pan-Tomkins algorithm [25], modified for R-peak correction and further smoothed using 4-beat ensemble averaging. Conventionally, computing AC features for red and IR PPG relies on robust peak and valley extraction. Although we were able to minimize respiratory artifacts through the breath-hold protocol, involuntary respiratory movements were still present and observable in some subjects. Evidently, extracting R robustly from respiration-corrupted PPG can be challenging [12,26]. Conventionally, the peak and valley of PPG in each cardiac cycle are extracted to compute the AC features [17] (pp. 129–130). In a preliminary analysis, we found that this method is not reliable as the signal can be easily distorted by the subtle—yet still significant—involuntary respiratory movements at this low perfusion site. To address this, we introduced a novel algorithm that does not require peak and valley extraction. Specifically, by rearranging the terms in Equation (1), we can obtain,
(4)R=ACredACIRDCredDCIR=ACred/IRDCredDCIR

By computing the AC_red/IR_, the ratio of AC_red_ to AC_IR_, we can avoid the difficulty of extracting peaks and valleys in distorted PPG signals. To do so, we leveraged the fact that the IR PPG beat, denoted as PPG_IR_, appears to have a similar morphology to the red PPG beat, denoted as PPG_red_, after being bandpass filtered. Therefore, we can model the relationship of the two PPG beats using a linear transformation method:(5)α1⋆,α2⋆=argminα1, α2∈ℝ‖PPGred−α1PPGIR−α2‖22
where PPGred,PPGIR∈ℝN, N is the number of samples in the PPG beat, and α1⋆,α2⋆ denote the scale and the bias that will minimize the ℓ^2^-norm of their differences. With the assumption that the differences, once optimized, should be closely distributed, we rejected beats with differences of more than five median absolute deviations from the median, which is a more robust rejection criterion compared to the “standard deviations around the mean” method [27]. Note we rejected only 1.79% using this method. The optimal scale, α1⋆, represents the ratio of the AC component of the two wavelengths:(6)ACred/IR=α1⋆

In parallel, the DC component was isolated using a low-pass filter with a high cutoff frequency at 0.1 Hz. This cutoff was based on a heuristic assumption that physiological dynamics of faster than 0.1 Hz (e.g., involuntary respiratory movement) do not directly relate to the deoxygenation induced by breath-hold based on data shown in [28,29,30]. The DC component was similarly segmented and smoothed to ensure consistency with the processing steps for AC extraction. Finally, DC features were computed as the mean of the segmented DC beats.

#### 2.4.4. PPG_green_-Based Outlier Rejection

Although we carefully selected the parameters of the preprocessing and feature extraction pipeline, some PPG beats may still be distorted due to motion artifacts and involuntary respiratory movements and therefore can hinder accurate SpO_2_ estimation. Hence, we designed a novel outlier rejection algorithm using the green PPG beats as a signal quality template for its robustness against noise [31], so as to reduce the contamination of abnormal features extracted. Our signal quality assessment relied on two assumptions: (1) reliable red or IR PPG beats in the bandwidth filtered constitute a morphology similar to that of green PPG beats; and (2) outliers in AC ratios are defined as datapoints that deviate by more than five median absolute deviations from the median (similar to the AC_red/IR_ rejection method). To determine the similarity, we consider a methodology described in [32]. First, the normalized cross-correlation (NCC) between a PPG beat with its corresponding template is computed:(7)$${\mathrm{NCC}}_{k,\;\lambda }=\frac{\sum_{n=1}^{N}\left({\mathrm{PPG}}_{\lambda }\left(n\right)-\bar{{\mathrm{PPG}}_{\lambda }}\right)\left({\mathrm{PPG}}_{\mathrm{green}}\left(n+k\right)-\bar{{\mathrm{PPG}}_{\mathrm{green}}}\right)}{\sqrt{\sum_{n=1}^{N}{\left({\mathrm{PPG}}_{\lambda }\left(n\right)-\bar{{\mathrm{PPG}}_{\lambda }}\right)}^{2}\sum_{n=1}^{N}{\left({\mathrm{PPG}}_{\mathrm{green}}\left(n+k\right)-\bar{{\mathrm{PPG}}_{\mathrm{green}}}\right)}^{2}}}\;$$
where PPG_λ_(n) denotes the n^th^ sample in the PPG beat of wavelength λ, $\bar{{\mathrm{PPG}}_{\lambda }}$ denotes the average value of the samples in a PPG beat of wavelength λ, and NCC_k, λ_ denotes the correlation coefficient between PPG_λ_ and the k-lag PPG_green_. Next, the maximal NCC_λ_, NCC_max,λ_, defined as
(8)$${\mathrm{NCC}}_{\max,\;\lambda }=\underset{k}{\mathrm{argmax}}{\mathrm{NCC}}_{k,\lambda }$$
is selected as a measure of the SQI of the PPG_λ_ and has a range of [0, 1]. Both assumptions translate directly to the two upper right blocks in Figure 4b. Each signal quality index (SQI) method has an empirically determined threshold, 0.7, and a sample is excluded if either SQI method suggests so. The PPG_green_-based outlier rejection algorithm rejected nearly 8.07% of the beats.

#### 2.4.5. Computation of R

The output matrix in Figure 5b has a dimension of N_beats_ × 5, where the five columns represent the AC features (AC_red/IR_), two DC features (DC_red_, DC_IR_), and the two binary SQI decisions (SQI_red_, SQI_IR_). Only features approved by the SQI algorithm were used to compute R. Note that we also experimented with the peak and valley method, but it would require a more aggressive outlier rejection threshold (~30% rejection ratio) to achieve comparable SpO_2_ estimation accuracy. Finally, R is computed by dividing AC_red/IR_ by $\frac{{\mathrm{DC}}_{\mathrm{red}}}{{\mathrm{DC}}_{\mathrm{IR}}}$ as shown below,
(9)R=ACred/IRDCredDCIR=α1⋆DCredDCIR 

In this dataset, R is a unit-less measure and generally ranges from 0.4 to 1.6 for SpO_2_ above 70%.

### 2.5. SpO_2_ Estimation

#### 2.5.1. Linear Regression

The temporally aligned SpO_2_ and the extracted R were subsequently used to train the parameters in Equation (3). The parameters *m* (slope) and *b* (intercept) were estimated by minimizing the ℓ^2^-norm of the difference between the ground truth SpO_2_ and estimated SpO_2_:(10)m⋆,b⋆=argminm, b ∈ℝ‖SpO2−mR−b‖22=f(x)
where x denotes pairs of SpO_2_ and R, and f represents an arbitrary function for determining the optimal parameters of an objective function.

#### 2.5.2. Training and Calibration Schemes

Since including a one-time, short calibration procedure is realistic for practical usage of the device, we also investigated the best training and subject-specific calibration procedure. Three training and calibration schemes were considered, including a (1) globalized scheme containing subject-independent training (see Figure 5a); (2) semi-globalized scheme featuring global training with subject-specific calibration (see Figure 5b); (3) subject-specific scheme (see Figure 5c). The globalized scheme is equivalent to the standard LOSO cross validation. The semi-globalized scheme described herein is analogous to the semi-globalized method discussed in [33], aside from the fact that we used duration rather than number of points to standardize the subject-specific calibration. Particularly, in the semi-globalized scheme the globally trained intercept *b* was replaced by a subject-specific calibrated *b* (using the data in the first of the 10 breath-holds). The subject-specific scheme involved training both parameters using only the first breath-hold data of the subject. Note that we also explored calibration using *m*, but the results were considerably worse and therefore not reported. In both globalized and semi-globalized schemes, LOSO cross validation was also used to assess generalizability of the models trained. To compare model performance fairly and to avoid data leakage, we excluded the first breath-hold of the test subjects for evaluation of the globalized schemes to ensure identical testing data.

#### 2.5.3. Evaluating Model Performance

To assess the performance of these three schemes, we recorded the RMSE, the parameters of the linear model on a per subject basis, and the Pearson correlation coefficient (PCC) of estimated SpO_2_ on all subjects jointly. The mean and the standard deviation of the subject-specific RMSEs were computed to summarize the performance of each scheme and subsequently used as the critical metric to assess the capability of the pulse oximetry. Note that the errors presented in this work are all absolute errors rather than relative/percentage errors. The unit of RMSE is denoted by %, which represent the oxygen saturation level.

## 3. Results

### 3.1. Accuracy of SpO_2_ Estimation

In Figure 6, regression plots and Bland–Altman plots are provided to demonstrate the estimation results. We also summarize the RMSEs across subjects, PCC, bias, and 95% limits of agreement (LOR) in Table 2. The globalized scheme achieves lowest accuracy (see Figure 6a,b). The semi-globalized scheme shows better accuracy (see Figure 6c,d). The subject-specific scheme achieves the best accuracy (see Figure 6e,f). Using the semi-globalized model, we were able to lower the mean RMSE by 0.36% and increase PCC by 0.02 when compared to the globalized model. The semi-globalized scheme and the subject-specific scheme have similar performance levels, both of which are superior to the globalized scheme. From the Bland–Altman plots, both models show minimal bias. 

### 3.2. Semi-Globalized Scheme vs. Subject-Specific Scheme

Since it has not been previously examined in the literature, we also studied which parameters benefit the most from subject-specific calibration. This is accomplished by comparing the semi-globalized scheme (i.e., calibrating *b*) to the subject-specific scheme (i.e., calibrating both *m* and *b*) while varying the calibration duration constraints. The duration constraint was imposed by considering data only within the said duration. Surprisingly, the semi-globalized model works more efficiently at reducing RMSE, as shown in Figure 7. Note 2 outlier subjects were excluded for better visualization. In all three duration constraints (10 s, 20 s, and 30 s), the semi-globalized schemes achieved a lower RMSE. When comparing the RMSE of calibrating b, constrained by a calibration duration of 10 s, to calibrating both parameters by a calibration duration of 30 s, we found no statistical significance (*p* > 0.05) as determined by a paired sample *t*-test. Therefore, we determined that the semi-globalized scheme is the best calibration strategy for this dataset as it would require a shorter duration to achieve similar performance to the subject-specific scheme.

### 3.3. Standardizing Subject-Specific Calibration: Duration vs. SpO_2_ Range

To study the most efficient way to collect data for calibration, we also examined the changes in RMSE by imposing different constraints on the calibration data, including a duration constraint and SpO_2_ range constraint. Similar to the duration constraint, the SpO_2_ range constraint considers data only within the said SpO_2_ range. According to the results shown in Figure 8a, we found that increasing calibration duration from 1 s to 20 s while fixing SpO_2_ range to 30% leads to significant (*p* < 0.05) reduction in mean RMSE across the subjects. On the other hand, the results shown in Figure 8b suggest that increasing the calibration SpO_2_ range from 1% to 20%, while fixing the duration to 30 s, did not lead to a significant (*p* > 0.05) difference in mean RMSE. Paired sample *t*-tests were used for the statistical analysis. Standardizing calibration duration appears to be the best calibration strategy here.

### 3.4. Effect of Varying Melanin Content

Since none of the subjects had nail polish on their right index finger or tattoos on their sternum, we only considered the confounding effect of the difference in melanin content. In this analysis, we assessed melanin levels using self-reported Fitzpatrick skin types [34] and studied the way melanin content affects the bias between finger and sternum SpO_2_. In Figure 9, the errors between different Fitzpatrick skin types are shown to be statistically insignificant (*p* > 0.05), using a one-way analysis of variance (ANOVA). This implies that our device does not introduce different bias for subjects with varying skin melanin content when compared to the finger pulse oximeter, understandably, as both operate on the same principles. However, it is worth noting that there are five subjects for skin type I, nine subjects for skin type II, three subjects for skin type III, three subjects for skin type IV, and zero subject for skin type V and VI. Due to the limited sample size and lack of data for the darkest Fitzpatrick skin types, the results attained here may not provide meaningful insight with a true accuracy of the proposed chest-based pulse oximetry on persons of all melanin levels. Note that in [35,36], it was reported that melanin content leads to SpO_2_ overestimation at low SaO_2_. Further investigation is required to study the way melanin content affects the accuracy of the chest-based pulse oximetry at various SaO_2_ levels.

## 4. Discussion

We unified previous evaluations of central-pulse oximetry and addressed relevant concerns while showing an accuracy that is comparable to the state-of-the-art [13]. To the best of our knowledge, this is the first thorough evaluation of chest-based pulse oximetry that jointly features a sufficient sample size, a wide dynamic range of SpO_2_, minimal respiratory artifacts, and rigorous cross validation to avoid data leakage. Furthermore, the study protocol, our alignment method, and key algorithmic components were described in full detail to allow for replication. This work paves the way for realizing the simultaneous monitoring of, in addition to SpO_2_, the cardiac, pulmonary, and cardiopulmonary functions using a small, standalone wearable patch device continuously and remotely, unlocking opportunities in personalized health intervention outside of a clinical setting.

### 4.1. Accurate SpO_2_ Estimation

We achieved low mean RMSEs for all training and calibration schemes, which were well within the criteria (RMSE ≤ 3.5%) for reflective pulse oximetry outlined by the FDA standard [16]. Our work addresses the challenges of the aforementioned approaches and estimated SpO_2_ accurately using a novel algorithm that proves to be robust, for PPG measured at this poorly perfused site. The breath-hold protocol successfully induced hypoxemia and reduced respiratory artifacts. Furthermore, the novel algorithms described in Section 2.4.2 to derive R leverage the morphological similarity between PPG_red_ and PPG_IR_. Our method avoids peak and valley extraction for distorted PPG beat, and proves to be less susceptible to artifacts. The PPG_green_-based outlier rejection algorithm was inspired by the robustness of PPG_green_ against motion artifacts [31]. Together, they alleviated difficulty in feature extraction for most PPG beats and excluded undesired PPG beats robustly.

### 4.2. Standardization of Subject-Specific Calibration

Besides accurate SpO_2_ measurements, we also designed experiments to identify the best training and calibration for this dataset for improving RMSE. More data points help to better calibrate the model to the test subject, and they do so by reducing the noise in the R extracted rather than capturing a wider SpO_2_ dynamic range, as evident from the results in Figure 6. Subject-specific calibration of *b* helps to reduce the randomness in the data. In contrast, if we were to calibrate *m* alone, we expect SpO_2_ range to have a more important role. Ultimately, we can benefit from both longer calibration duration and wider SpO_2_ range, but in a situation where hypoxemia is not preferred by the intended users, we have shown that limited SpO_2_ dynamic range with a sufficiently long calibration duration can still efficiently improve accuracy. This finding is beneficial for the usage of the device since it alleviates the need to induce changes in SpO_2_ and consequently makes the subject-specific calibrating procedure more practical and safer. Consequently, if necessary, breath-hold duration should be standardized instead of the SpO_2_ range.

When observing the data used to calibrate both *m* and *b* for test subjects, we noticed that the first breath-hold is consistently shorter across subjects. As a direct consequence, data of the first breath-hold generally does not have a wide SpO_2_ dynamic range. Furthermore, subjects seemed to be able to hold their breath longer due their adaptive tolerance to withstand the vaguely defined “discomfort” [37]. Since estimating the slope *m* requires a sufficient dynamic range of SpO_2_, calibrating *m* using just the first breath-hold is usually not enough. Using all 10 breath-holds and adequate SpO_2_ dynamic range across all training subjects offers a clear advantage to the globally trained *m* over the calibrated *m*. Hence, when the calibration data were limited (within the duration of one breath-hold), training *m* globally and calibrating *b* using the test subject’s data can achieve better accuracy.

### 4.3. Practical Use Case

The current manufacturing cost of the device is on the order of $200. However, producing this device on a large scale can reduce the cost substantially as components would be ordered in volume and manufacturing processes can be refined to improve manufacturability. We do not foresee any challenges with scalability as the devices manufactured so far show robust functionality. The practical subject-specific calibration procedure can be designed by aggregating the conclusions made thus far. We suggest a 15 s breath-hold during which the ground truth SpO_2_ and biosensor data are collected from a target subject. This breath-hold duration is selected because all breath-hold durations across the subjects in this study exceed 15 s. Using the data from the subjects analyzed in this study, we found the globally trained slope *m*_global_ to be −21.54 and the intercept *b*_global_ to be 106.69. Following the semi-globalized scheme, *b*_global_ can be replaced by *b*_subject-specific_, which was calibrated using data from the 15 s breath-hold of the target subject. The resulting subject-specific linear model takes the form of SpO_2_ = *m*_global_ × R + *b*_subject-specific_. One potential reason for calibration failure could be the adoption of smoking behavior. According to [38], smokers have elevated levels of carboxyhemoglobin (COHb). As a result, assumptions (i.e., the only hemoglobin species in the arteries are Hb and HbO_2_) made in Equation (2) are violated, which can subsequently lead to the overestimation of SpO_2_. 

### 4.4. Limitations

One key limitation is that we only validated the accuracy of our pulse oximetry during segments with minimal respiratory artifacts. Kramer et al. [13] previously reported high accuracy from subjects undergoing spontaneous breathing despite a lack of details of their algorithm. Future work should investigate whether our proposed ACr_ed/IR_ extraction and PPG_green_ outlier rejection algorithm can withstand respiratory artifacts more severe than the involuntary respiratory movements during breath-hold and attain similar accuracy. In addition, we noticed that the deoxygenation dynamics of R still may not be perfectly aligned to those in SpO_2_, even after precise alignment. Our alignment method assumed that the delay between the start of breath-hold and chest SaO_2_ deoxygenation is distributed across the integer values in the interval [−10, 10], as described in Appendix A. However, the delay may exceed beyond this, and therefore it is likely that we still captured an undesired delay for some subjects. Specifically, inter-subject and intra-subject variability in this delay may directly translate to variability in the estimated *b*, which explains the importance of calibrating *b*. For example, consider the case where the error in alignment is 1 s in a breath-hold of a subject. The desaturation rate, of around 0.26% per second, can be roughly estimated from Figure 3. The 1 s error in alignment can lead to a difference of 0.26% for all datapoints of that breath-hold, and therefore, systematically introduces a bias. Besides this bias, the breath-hold study method may also lead to another shortcoming. The SpO_2_ at different measurement sites may not map to one another completely, even if the delay has been accounted for [5]. This is understandable because the sternum and finger have different tissue and vascularization compositions and therefore deliver and consume oxygen at different rates as well. Finally, similar to most commercial finger pulse oximeters, the chest-based pulse oximetry may suffer from motion artifacts, the presence of Hb_CO_ and Hb_MET_, venous pulsation [39], etc. However, we expect to considerably improve RMSE further if we can induce deoxygenation slowly and compare the estimation when using the validation protocol suggested by the FDA [16] and show improved accuracy of biosensor’s measurements despite the above limitation. 

### 4.5. Future Work

The results and techniques demonstrated in this work allow for the accurate measurements of SpO_2_, which can in turn be used to better inform underlying pulmonary dysfunctions unobtrusively, continuously, and remotely. Together, with its ability to measure cardiac function, we can next validate the wearable patch biosensor for its ability to quantitatively and objectively assess disease progression of cardiovascular and pulmonary diseases such as COVID-19, nocturnal hypoxia caused by sleep apnea, and high-altitude sickness. Ultimately, tracking these health parameters may provide a better understanding of the cardiopulmonary-related comorbidities and consequently facilitate the adoption of longitudinal wearable monitoring devices, for detecting underlying disease when symptoms are subtle and unnoticeable.

## 5. Conclusions

Here, we demonstrated that our custom, chest-worn wearable patch biosensor was capable of accurately estimating SpO_2_ while subjects underwent a 10 breath-hold protocol. We presented that standardizing the calibration duration, rather than calibration range, was the most important factor for optimal calibration. Finally, we found that differences in Fitzpatrick skin types do not introduce disparities in bias. Future studies will focus on improving the study protocol to induce gradual changes in SaO_2_ as per the FDA guidelines, while measuring gold standard SaO_2_ simultaneously through a co-oximetry of arterial blood samples [16], designing algorithms that mitigate respiratory artifacts when present, and by recruiting a larger population that is demographically diverse, especially participants with higher Fitzpatrick skin types. Together with its holistic cardiac monitoring, this device can provide longitudinal and quantitative information of disease progression in both cardiovascular and pulmonary diseases.

## Figures and Tables

**Figure 1 biosensors-11-00521-f001:**
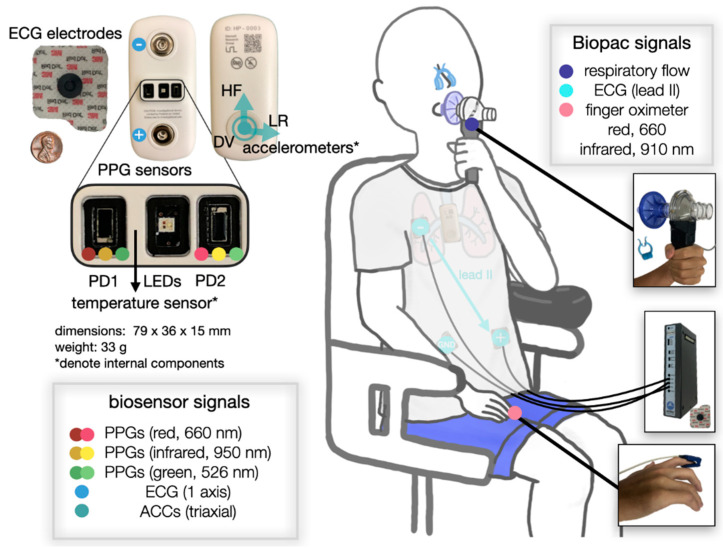
Illustration of a subject undergoing the breath-hold study. Placements of the wearable patch biosensor (**left** side) and the ground truth Biopac sensors (**right** side) are depicted. The relative size of the biosensor is shown with an off-the-shelf ECG electrode and a penny. Note that the photodiode (PD) has an area of 4.5 mm^2^. PD1 is on the left side and PD2 is on the right side of the subject.

**Figure 2 biosensors-11-00521-f002:**
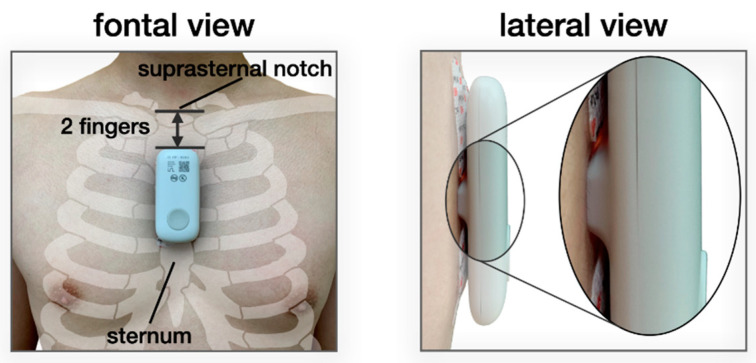
From the frontal view, the wearable patch biosensor is attached to the sternum of the subject using ECG electrodes. The superior end of the device starts approximately two fingers down from the suprasternal notch. From the lateral view, the protruded part of the device that houses the LEDs and PDs is visible. Note that light is hardly visible from the sides of the device.

**Figure 3 biosensors-11-00521-f003:**
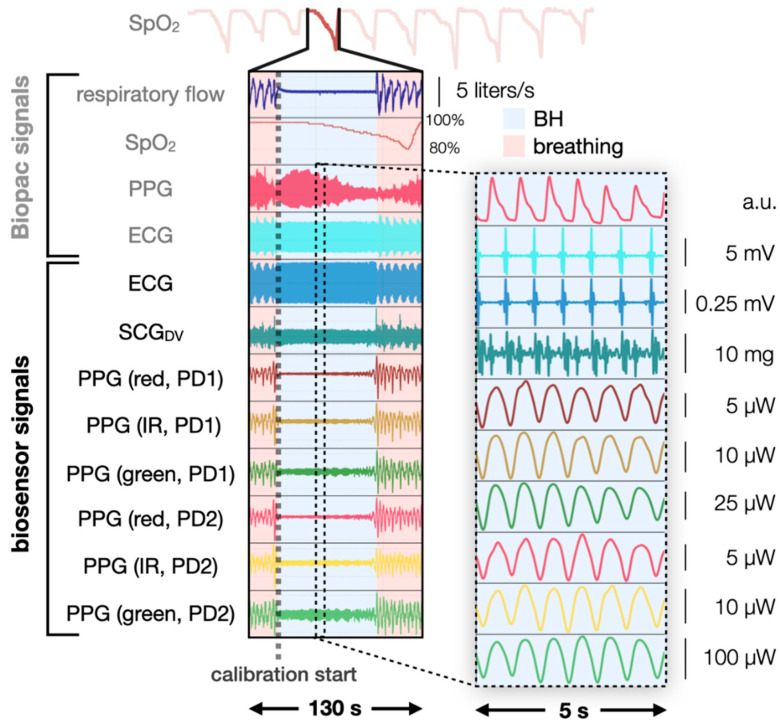
Biopac and biosensor signals during the fourth breath-hold (BH, blue) and during normal breathing (pink). A 5 s window during breath-hold is shown to illustrate the quality of the signals.

**Figure 4 biosensors-11-00521-f004:**
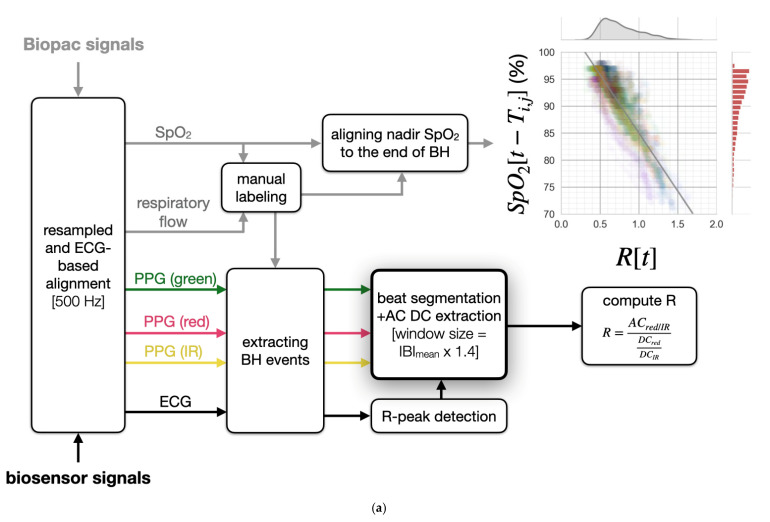
Feature extraction block diagrams. (**a**) Overview of the signal preprocessing and manual labeling pipeline. Extracted pairs of ratio of ratios (R) and SpO_2_ were used for training and calibration of the model. *T_i,j_* denotes the delay found between R and SpO_2_ for the *j*th breath-hold of subject *i*. The aligned data were displayed in the upper right scatter plot. The distribution of R and SpO_2_ are also shown above and on the right of the scatter plot, respectively. IBI_mean_ denotes the mean of the interbeat intervals (IBI) of a subject. (**b**) The block diagram for PPG beats segmentation, AC_red/IR_ extraction, DC feature extraction, and PPG_green_-based outlier rejection.

**Figure 5 biosensors-11-00521-f005:**
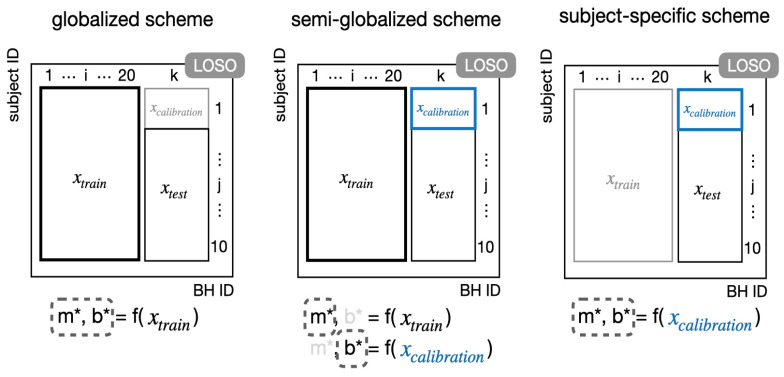
Different model training schemes, where x_train_ denotes the data from the training subjects; x_calibration_ denotes the first breath-hold data from the test subject; x_test_ denotes the data of the test subject excluding the first breath-hold; k denotes the left-out subject. The parameters used to construct the model are in the dashed boxes. Data or parameters not used at all were colored in light gray. (**a**) Globalized scheme (no subject-specific calibration). (**b**) Semi-globalized scheme (with subject-specific calibration). (**c**) Subject-specific scheme.

**Figure 6 biosensors-11-00521-f006:**
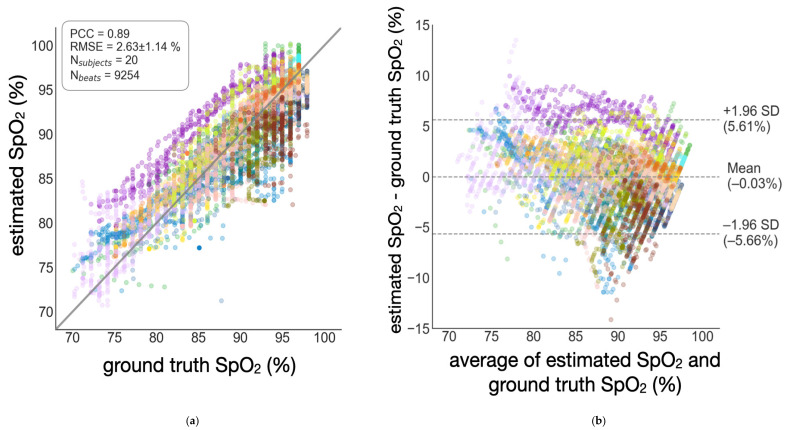
Estimation performance of the wearable patch biosensor. The root-mean-square error (RMSE) across subjects and Pearson correlation coefficient (PCC) are shown. Note each color represents a subject. (**a**) Regression analysis of the globalized scheme. (**b**) Bland–Altman analysis of the globalized scheme. (**c**) Regression analysis of the semi-globalized scheme. (**d**) Bland–Altman analysis of the semi-globalized scheme. (**e**) Regression analysis of the subject-specific scheme. (**f**) Bland–Altman analysis of the subject-specific scheme.

**Figure 7 biosensors-11-00521-f007:**
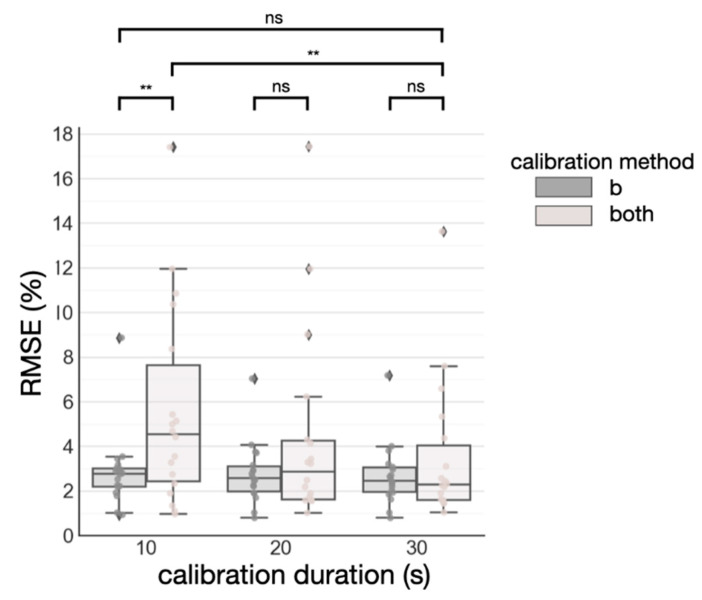
Calibration using *b* and using both *b* and *m*. *ns* denotes not significant (*p* > 0.05), and a double asterisk denotes significant differences (*p* < 0.01) as determined by paired sample *t*-tests.

**Figure 8 biosensors-11-00521-f008:**
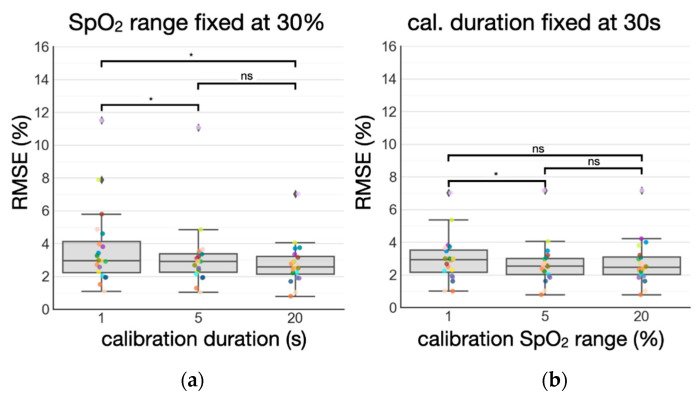
Calibration analysis. (**a**) RMSE is reduced significantly with increasing calibration duration while fixing SpO_2_ range. (**b**) RMSE does not change significantly with increasing SpO_2_ range while fixing calibration duration. *ns* denotes not significant (*p* > 0.05), and a single asterisk denotes significant differences (*p* < 0.05) as determined by paired sample *t*-tests.

**Figure 9 biosensors-11-00521-f009:**
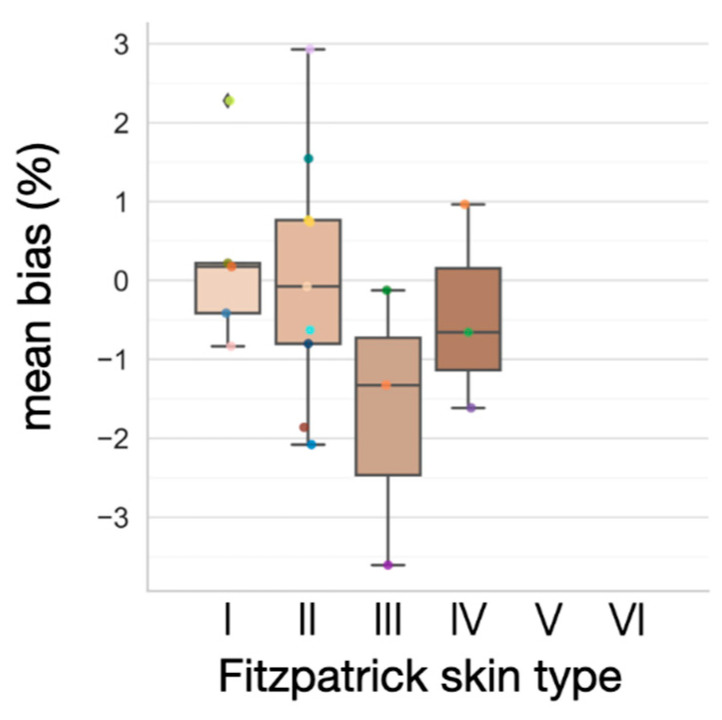
Mean bias across varying Fitzpatrick skin types. The differences between Fitzpatrick skin types are not statistically significant (*p* > 0.05). One-way analysis of variance (ANOVA) was used for statistical analysis.

**Table 1 biosensors-11-00521-t001:** Demographics and physiological responses during breath-hold of the subjects included in the analysis.

Variable		Female (N = 6)	Male (N = 14)
Demographics	Age (years)	28.00 (2.00)	26.15 (2.19)
	Fitzpatrick skin type (I–VI)	1.83 (1.17)	2.36 (0.93)
	Weight (kg)	56.35 (11.61)	76.44 (11.97)
	Height (cm)	163.03 (7.46)	178.03 (7.09)
	BMI (kg/m^2^)	21.02 (2.86)	24.05 (2.84)
Breath-hold response	Baseline SpO_2_ (%)	96.50 (0.84)	96.29 (1.27)
	Nadir SpO_2_ (%)	88.80 (4.81)	80.52 (8.70)
	Breath-hold duration (s)	44.07 (25.64)	55.99 (15.52)
	Approximate finger SpO_2_ delay (s)	24.91 (8.69)	27.42 (12.49)
Perfusion index (PI)	Red PPG (%)	0.05 (0.03)	0.07 (0.04)
	Infrared PPG (%)	0.08 (0.04)	0.10 (0.06)
	Green PPG (%)	0.56 (0.23)	0.60 (0.40)

Note values are presented in mean (standard deviation).

**Table 2 biosensors-11-00521-t002:** Comparing the results of three training schemes based on their RMSE, PCC, bias, and 95% LOR.

	RMSE	PCC	bias	95% LOR
Globalized	2.63 ± 1.14%	0.89	−0.03%	[−5.66%, 5.61%]
Semi-globalized	2.27 ± 0.76%	0.91	0.11%	[−4.81%, 5.02%]
Subject-specific	2.27 ± 0.88%	0.92	0.24%	[−4.84%, 5.31%]

## Data Availability

The data presented in this study are available on request from the corresponding author. The data are not publicly available due to them containing information that may compromise the privacy of the research subjects.

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
