# Peer review of "Enabling Continuous Wearable Reflectance Pulse Oximetry at the Sternum"

_biosensors, 2021, doi:10.3390/bios11120521_

Round 1
Reviewer 1 Report
This is an interesting paper reporting a chest-based wearable patch biosensor for recording some physiological signals like ECG and PPG. To find the relationship between R and , they developed a feature extraction algorithm based on linear transformation and an outlier rejection algorithm based on . A linear regression model was used to estimate the from the extracted R. The feasibility and accuracy of their sensor and algorithms were well validated by a breath-hold experiment they designed themselves. Overall, this manuscript is well-written and is recommended to be published in Biosensors after addressing my comments below.
- PPG is one of the noninvasive methods to detect changes in blood volume through photoelectric means. the authors should introduce the principle of this method before experiment.
- In the experiment, the authors used two symmetric photodiodes to receive the reflected signals and selected by visual inspection. So, ensuring the deviation between the two photodiodes is important.
- All the subjects involved in the experiments were young. Will the accuracy of the estimation algorithm be affected when the subjects included the other age groups?
- Some detailed information on the materials and equipment used in the experiment are missing.
Author Response
Dear Reviewer,
We thank you for your careful and insightful review of our work. Please see the attachment for our point-by-point responses.
Sincerely,
Michael Chan

Reviewer 2 Report
The authors propose a new means of measuring long-term blood oxygen monitoring. The paper is interesting and combines a number of new techniques. My only main requirement is to understand more about the device being used to collect the data. The paper is a little over-written and it could have been written in a more concise way. I only have a few comments, which are below.
Introduction: I don’t like the structure of the introduction. I would prefer it didn’t have sections within it and it was shorter and clear what is novel about the work. The other sections should go in section 2 where there is discussion on the methods. In addition, please remove the results from the introduction. Just be clear about what is novel about the work (so section 1.4). Also, some of this should go in the discussion later.
Section 2: I would like to see some text section before going into a large table. It makes the paper very odd…
Line 128: Breath hold study should be described in more detail (for example exactly what it is…). Or this may be later.
Line 156: I would emphasis here that this is your new device. I would also like to see some technical information regarding the unit (design specification, communication protocols etc.). If this is in another paper, please just give an overview of what is inside the unit and then a reference.
Line 217 onwards: Please ensure all parameters in the equations are defined.
Line 371: May be better to tabulate the results.
Lin 387: Colours not defined in figures
Line 395: Most of the figure legend should go in the text.
Line 499: How long will the calibration hold for and how rapid is the loss of accuracy to an unacceptable level?
Line 535: I think some discussion about cost and scalability is needed within the discussion. Then what are the next steps of the work.
Conclusions: I would reduce this length and move the new statements into the discussion
Author Response

(The authors gave the same response as above.)
